# MUG: INTERACTIVE MULTIMODAL GROUNDING ON USER INTERFACES

## ABSTRACT

We present MUG, a novel interactive task for multimodal grounding where a user and an agent work collaboratively on an interface screen. Prior works modeled multimodal UI grounding in one round: the user gives a command and the agent responds to the command. Yet, in a realistic scenario, a user command can be ambiguous when the target action is inherently difficult to articulate in natural language. MUG allows multiple rounds of interactions such that upon seeing the agent responses, the user can give further commands for the agent to refine or even *correct* its actions. Such interaction is critical for improving grounding performances in real-world use cases. To investigate the problem, we create a new dataset that consists of $77,820$ sequences of human user-agent interaction on mobile interfaces in which $20\%$ involves multiple rounds of interactions. To establish our benchmark, we experiment with a range of modeling variants and evaluation strategies, including both offline and online evaluation—the online strategy consists of both human evaluation and automatic with simulators. Our experiments show that allowing iterative interaction significantly improves the absolute task completion by 18% over the entire test set and 31% over the challenging subset. Our results lay the foundation for further investigation of the problem.

## 1 INTRODUCTION

Natural language understanding on graphical user interfaces (GUIs) is crucial for realizing human-computer interaction and assisting scenarios that have accessibility difficulties (Sarsenbayeva, 2018). Specifically, interpreting user commands into executable actions has drawn increasing interests as it manifests rich research problems including multimodal modeling and natural language grounding (e.g., Li et al., 2017; Gur et al., 2019; He et al., 2020; Li et al., 2020a; 2021). Prior works often consider UI grounding in a single-pass fashion where the model predicts actions with a given instruction without looking backward to refine prediction. However, in a realistic scenario, user instructions can be ambiguous or inaccurate especially when the target action is difficult or inconvenient to articulate. Reasoning in such cases is inherently iterative. Therefore, it is important and beneficial to incorporate interaction for resilient grounding (Suhr et al., 2019; Chandu et al., 2021).

In this paper, we investigate interactive grounding on GUIs, which aligns multimodal input to actionable objects of a screen. We focus on single-screen interaction which is the building block of UI reasoning. Specifically, we introduce the MUG (**M**ulti-turn **UI G**rounding) task in which the user iteratively guides the agent to select a desired UI object (see Fig. 1). With a given UI and a target object, the user instructs the agent via natural language, ranging from casual intent to more descriptive commands. The agent infers which UI object is intended by the user and and highlights it. If the agent is correct, the user can confirm the selection and the grounding is completed. Otherwise, the user issues further guidance, e.g., *"Click the one below"*, to the agent to refine its selection. We collecte the MUG dataset from live interaction sessions between pairs of human annotators—one acts as the user and the other as the agent. Our dataset has $77,820$ examples, each records the transaction history in a session. Specially, $20\%$ of the dataset are challenging ones as their human commands need multiple rounds to ground, even for human agents.

To establish the benchmark, we experiment with a range of variants to model the dynamics between the two roles. While the main goal of the task is to develop agent models for grounding, we also develop the user models for online instruction simulation. We build our models upon a Transformer-

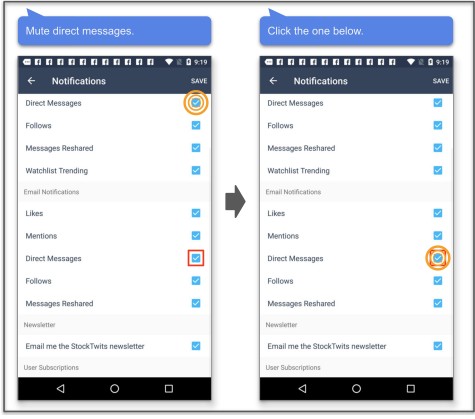
(a) Example one.

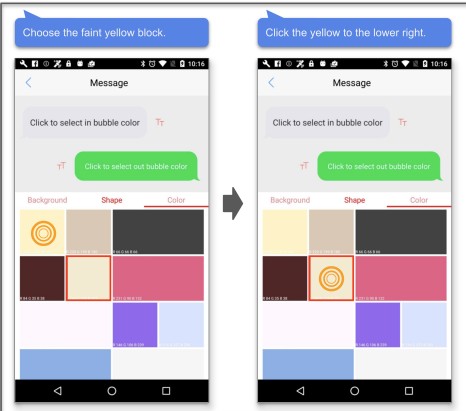
(b) Example two.

Figure 1: Two illustrative examples of the MUG task. There are two turns in each of these examples. Interactions happen within a single screen. User commands are shown above the screens. The target object is bounded in ▢. Agent choices are marked with ◎.

based encoder-decoder architecture (Li et al., 2021), and experiment with various learning methods, including traditional sequence modeling and reinforcement learning. To fully examine the model performances, we evaluate the agent model with a spectrum of evaluation strategies, including both offline and online evaluations. For the online evaluation, we employ both automatic and human evaluations, which include interactions between the agent and the user (either a human or the user model) and offer a comprehensive probe into model understanding. Our experiments show that incorporating interaction substantially improves UI grounding task completion by $18\%$ on the entire dataset and $31\%$ on the challenging set, both in absolute scales. Furthermore, our robustness measurements suggest MUG, while being a seemingly easy single-screen task, is actually difficult since neural agents sometimes struggle to correct themselves, resulting in repeated wrong selections across multiple turns. This suggests large rooms for future improvement in grounding agents.

In summary, our key contributions[1] are:

1. We introduce MUG, a novel interactive vision-language task that focuses on multi-turn language grounding on a graphical UI screen, which is a challenging task that is meant to improve language grounding in realistic UIs.

2. We create a rich dataset that includes 77,820 examples recorded from live sessions between pairs of human users and agents. And 20% of the data are challenging for both human annotators and neural agents.

3. We experiment with a range of model variants and evaluation strategies, showing that iterative interaction significantly improves grounding accuracy by $18\%$ and $31\%$ on the entire and challenging test sets respectively, with automatic assistance from our user models. Our work lays a solid foundation for future investigations on collaborative grounding.

## 2 BACKGROUND

Multi-modal modeling has a long history of research (e.g., Winograd, 1972; Barnard & Forsyth, 2001; Lavrenko et al., 2003; Plummer et al., 2015; Yu et al., 2016). One important area focuses on grounding objects in images where the natural language is used as an additional input (Chen et al., 2017; Yu et al., 2016; 2018; Fukui et al., 2016; Deng et al., 2021).

**Interactive Multimodal Grounding** Prior works have formulated grounding as a multi-step reasoning task, e.g., navigation via multiple steps of grounding (e.g., Ku et al., 2020; Gur et al., 2019). Our work differs by focusing on agent's ability to self-correct in synchronized turns of interaction

---

[1]The dataset and code for reproducing our experiments are at https://github.com/to-be-de-anonymized.

on a UI screen. It is also conceptually linked to repeated reference game (Hawkins et al., 2020), except we use a different form of communication (language-action) instead of dialogue (language-language). Our task leverages iteratively refined instructions on atomic action instead of the increased instruction utility over multi-step actions (Effenberger et al., 2021). Our work models both the user and the agent, and let them communicate online. This is different from single-sided modelings (Suhr et al., 2019; Kojima et al., 2021). Our observation that interaction improves grounding is also in line with dialogue-based works (e.g., Haber et al., 2019; Takmaz et al., 2020).

**UI Grounding**   Grounding UI objects involves automatic completion of actions on web or mobile interfaces (e.g. Pasupat et al., 2018; Li et al., 2020a; He et al., 2020). It is also an important accessibility task for users who are situationally impaired when they are occupied by real-world tasks at hand (Sarsenbayeva, 2018). Compared to grounding on natural images, these tasks usually take well-specified user commands and aim to select the object that best matches the command. The UI image is often encoded via ResNet (He et al., 2016) or ViT (Dosovitskiy et al., 2020). The structure and text features of UI are often encoded by Transformer model (Vaswani et al., 2017). Fusing multimodal information is widely handled by cross-attention (e.g. He et al., 2020; Li et al., 2021; Bai et al., 2021). We adopt these neural components in our benchmark.

**Mobile UI Datasets**   Many grounding tasks, while covering multiple screens, remain one-pass reasoning, such as PIXELHELP (Li et al., 2020a) and MoTIF (Burns et al., 2022). Prior works (e.g., Todi et al., 2021) model sequence of (action, state) pairs via reinforcement learning (RL). In contrast, MUG focuses on correcting a single action on one screen. Tab. 1 summarizes key differences among other Mobile UI datasets. Importantly, MUG is a challenging task as it enables corrective interaction in synchronized turn between user and agent.

| Data | Screen | Instruction | Natural Instruction | Interactive Correction |
|---|---|---|---|---|
| RICO (Deka et al., 2017) | multi | ✗ | ✗ | ✗ |
| PIXELHELP (Li et al., 2020a) | multi | ✓ | ✓ | ✗ |
| MoTIF (Burns et al., 2022) | multi | ✓ | ✓ | ✗ |
| RICOSCA (Li et al., 2020a) | single | ✓ | ✗ | ✗ |
| REFEXP (Bai et al., 2021) | single | ✓ | ✓ | ✗ |
| MUG (Ours) | single | ✓ | ✓ | ✓ |

Table 1: Comparison to prior mobile UI Datasets.

## 3   TASK FORMULATIONS

As a grounding task, MUG involves two participants: a user and an agent. Our formulation includes both roles to provide a holistic view of interactive grounding. The user's goal is to instruct, via natural language, the agent to select the desired object $g$ on the UI screen $S$. The unique aspect of MUG is that it allows the user to guide the agent *iteratively* to identify the target action by issuing a series of commands, each in response to the agent's prior inference.

We separate such user-agent interaction into turns. At turn $t$, the interaction consists of:

$$\begin{cases} c_t : \text{user command,} \\ a_t : \text{agent action.} \end{cases}$$

where the user first instructs the agent with command $c_t$, and the agent responds with a suggestion of action $a_t$. Here $a_t$ is essentially the index of object. The task is completed when $a_t = g$.

### 3.1   AGENT TASK

In MUG, the action space for the agent consists of a set of UI objects to click on the interface, e.g., in Fig 1. Intuitively, we would want the agent to take the desired action $g$ as early as possible, i.e., as few turns as possible. Thus, at turn $t$, the agent models

$$P_\theta(a_t | S, c_{[0,t]}, a_{[0,t-1]}) \tag{1}$$

where $\theta$ denotes the agent parameters. This iterative grounding early stops once $a_t = g$ or $t$ reaches a maximum number of turns allowed.

## 3.2 USER TASK

The user's role is to provide guidance to the agent through iteratively refined instructions. In contrast to one-pass prediction tasks (e.g. Pasupat et al., 2018; He et al., 2020) where the agent makes a one-shot guess, a MUG user issues follow-up commands that are dependent of prior instructions $c_{[0,t-1]}$ and agent actions $a_{[0,t-1]}$, which is formalized as the following:

$$P_\phi(c_t|S, g, c_{[0,t-1]}, a_{[0,t-1]}) \tag{2}$$

where $\phi$ denotes the user. Here, the user model is aware of the target object $g$.

**Interplay between User and Agent**   The agent task (Eq. 1) is the pivot of MUG. The user task (Eq. 2) aims to guide agent towards task completion, which potentially includes online training. In our benchmark, we let the user and agent play together. Although automatic evaluation is not as realistic as human evaluation, it offers a fast, low-cost, and reproducible environment. This setting also allows us to study various questions surrounding the interplay between the two, e.g., whether an automatic user can assist an agent? and whether agent errors would confuse the user?

## 4 DATASET CREATION

As there is no available dataset for model training and evaluation, we developed an interactive labeling interface to collect data for MUG. Our data collection involves two human annotaters to play the roles of the user and the agent respectively in a live session. The user and the agent have two separate views, running on different machines (Appx. A). Both views share the same UI screen and a message box showing instruction history. Our task embodies the *eyes-on, hands-free* situation for mobile interaction where the user is required to only use language for the task, and the machine responds its prediction by highlighting. The user can commit the action if the prediction is confirmed. During a session, only the user can see the target; and the the message box is read-only to the agent thus no language-base dialogue would take place.

## 4.1 ANNOTATION WORKFLOW

We use the UI corpus, mobile UI screenshots and view hierarchies, from RICO (Deka et al., 2017) and auxiliary object features from the CLAY dataset (Li et al., 2022). Each session starts with a randomly sampled UI object (e.g., a *button*), from the visible view hierarchy, as the target object $g$. User annotators are encouraged to articulate their initial command ($c_0$) casually or goal-oriented. We consider such design to cover the realistic scenarios discussed in Sec. 1, and potentially free users from composing long and precise instructions.

In the agent view, all clickable objects on the UI screen are revealed with their bounding boxes highlighted, which show what objects the agent can select, without indicating which one is the target $g$. The current agent selection is reflected on both the user and the agent's view. The session continues to the user's turn if the agent selection does not match $g$. In follow-up turns, the user is not allowed to repeat a command issued in previous turns, and likewise the agent is not allowed to select an previously chosen object. Upon the agent selection matching the target object in the user view the task is completed. Each session allows up to $5$ turns and we filter out those unfinished.

## 4.2 RESULTS & ANALYSIS

We collected 77,820 examples based on 31,265 unique screens from 7,132 apps (see details in Table 2). We split the dataset into the training, development, and test sets. We use app-wise split (Li et al., 2020b) to avoid potential leaking across sets. As shown in Table 2, the three splits have a similar distribution regarding the number of turns in each example.

In general, users tend to provide short instructions ($\sim 4$ words). Across the dataset, annotators completed $\sim 80\%$ examples in one turn ( i.e., one instruction and one selection ). The rest $20\%$

| | | Statistics of examples | | | | Distribution of Turns (%) | | | | |
|---|---|---|---|---|---|---|---|---|---|---|
| Split | Apps | Screens | Interactions | Avg. #Turns | Avg. #Token/Turn | 1 | 2 | 3 | 4 | 5 |
| Train | 6,039 | 26,090 | 65,235 | 1.24 | 4.26 | 78.91 | 18.31 | 2.37 | 0.35 | 0.06 |
| Dev | 544 | 2,625 | 6,377 | 1.23 | 4.18 | 79.99 | 17.77 | 1.91 | 0.27 | 0.06 |
| Test | 549 | 2,550 | 6,208 | 1.23 | 4.18 | 80.20 | 16.82 | 2.55 | 0.40 | 0.03 |
| All | 7,132 | 31,265 | 77,820 | 1.24 | 4.25 | 79.10 | 18.15 | 2.35 | 0.35 | 0.06 |

Table 2: Dataset statistics.

dataset is considered to be more challenging, with the majority taking $2 - 3$ turns. We call this $20\%$ as the *Challenging* subset. In Sec. 6, we will show that examples requiring more turns for human agents are also more difficult for grounding models. We refer readers to Appx. F for examples.

We should note that while the $20\%$ multi-turn ratio seems a low percentage but it leads to large impact in practice. Firstly, it lines up an upper bound for agent ground accuracy (i.e., $80\%$). Moreover, real-world grounding sessions can also span multiple screens. If we apply the same $20\%$ rate here, the probability for a multi-screen session to require correction is much more significant. For instance, $67\%$ with 5 screens. Lastly, we will see in Sec. 6 that neural agents could also struggle at instructions that are trivial for human to execute.

In Table 3, we categorize 200 Challenging examples from the development split. We found followup commands are mainly to make spatial adjustments ($50\% + 10\%$) or to add extra information.

| Percentage | Attribution | Example |
|---|---|---|
| 50% | Adjusting relative position in the layout. | *the value before the text.* |
| 31% | Providing more information of the target. | *show me channels. $\rightarrow$ click tv icon.* |
| 10% | Adjusting direction/position on the screen. | *not reward but collect at the bottom.* |
| 3% | Rephrasing the instruction. | *go to books. $\rightarrow$ show me books logo.* |

Table 3: Major categories for the second turn from 200 examples in the development split.

## 5 GROUNDING MODELS

We aim to have a general architecture for the UI domain and explore its variants to model multi-turn interaction. Our agent model is based on a transformer encoder-decoder network, inspired by (Li et al., 2021). The encoder comprehends the objects on the UI screen (Sec. 5.1) and the decoder predicts a target action based on the UI and interaction history (Sec. 5.2).

### 5.1 UI ENCODER

Our encoder processes the interface $S$. Each $S$ consists of two modalities of information, i.e., a screenshot $I_S$ and view hierarchy features $\psi$ (Deka et al., 2017; Li et al., 2022). The concrete list of $\psi$ is in Appx. D. The output is an encoding $v^k$ for each object indexed by $k$, similar to (e.g., Li et al., 2020a; He et al., 2020; Li et al., 2020b):

$$\Phi_S = \text{ResNet}(I_S) \tag{3}$$

$$v = T_{\text{enc}}(\{\text{ROI}^k(\Phi_S)|\psi^k\}) \tag{4}$$

For the image, we use a pre-trained ResNet-50 (He et al., 2016) which is fine-tuned with other modules. The resulted $\Phi_S$ (grid size of $h \times w$) is then mapped to object level by region-of-interest (ROI) pooling (Ren et al., 2015). Finally, we fuse multimodal features for each object by a transformer encoder $T_{\text{enc}}$. The output $v$ stands for a sequence of object encodings which are interaction-agnostic.

### 5.2 GROUNDING DECODER

We use a causal transformer $T_{\text{dec}}$ to predict from interaction history. The architecture is similar to (Li et al., 2021), except we use multi-turn interaction as input, e.g., $c_{[0,t]}$ and $a_{[0,t-1]}$. The output of $T_{\text{dec}}$

is a vector $z_t$ that summarizes prior interaction up to $c_t$:

$$z_t = T_{\text{dec}}(v, c_0, v^{a_0}, c_1, ..., v^{a_{t-1}}, c_t) \tag{5}$$

where $a_t$ denotes object index, either from model prediction or human selection. The specific input to Eq. 5 will be subject to modeling variants in Sec. 6.1. For classification, we use a linear layer $f$ to score the $k$-th object:

$$a_t = \arg \max_k f([z_t|v^k]) \tag{6}$$

## 6 EXPERIMENTS

The goal of our experiments is to explore training and evaluation methods for MUG and establish a benchmark. Simply matching the initial instruction and object text is insufficient as text features are often incomplete. For instance, the validation split has $46\%$ objects missing text, and a deterministic classifier using METEOR (Banerjee & Lavie, 2005) has only $21\%$ F1. In this section, we present setups for the neural architecture discussed in Sec. 5. In Sec. 6.1, we explore multiple variants for the agent. For automatic evaluation, we present a simple and effective heuristics-based user model and a neural version in Sec. 6.2,. Finally, we show extensive F1 results in Sec. 6.4 and 6.5, robustness in 6.6, ablations in 6.7 and 6.8, and error analysis in 6.9. We refer readers to Appx. E for hyperparameters, Appx. H for sample predictions, and Appx. G for extended discussion.

To avoid confusion, we thereafter use $a'_t$ to refer to the selection predicted by the agent model at turn $t$, while $a_t$ to the human agent's selection. Similarly, we refer $c'_t$ to instruction generated by the user model while $c_t$ to the one from the human user.

### 6.1 AGENT MODELS

Our agent models use the $T_{\text{enc}}$ and $T_{\text{dec}}$ (in Sec. 5) as a backbone, denoted as $\theta$. Recall that $T_{\text{enc}}$ processes $S$ while $T_{\text{dec}}$ processes interaction. Here, we discuss different handlings of $T_{\text{dec}}$.

**Single or Multi-turn Model**   The first factor we investigate is how allowing multiple turns helps grounding. For each example, we can feed the entire interaction history as input to the agent model and supervise agent selection on the very last turn $T$:

$$P_\theta(a'_T = g|S, c_{[0,T]}, a_{[0,T-1]}) \tag{7}$$

We can further reduce the input to be $(S, c_0)$ only, making a single-turn model. To evaluate single-turn model with multi-turn examples, we simply concatenate all $c_t$ into one instruction.

**Instruction-only Model**   To understand how it helps grounding by taking into account of previous actions of the agent in the multi-turn model (Eq. 7), we introduce the command-only baseline, which ignores agent actions (selections) in the interaction history:

$$P_\theta(a'_T = g|S, c_{[0,T]}) \tag{8}$$

**Imitation Model**   Instead of supervising the agent only at the last turn, we can model the entire action sequence as an imitation model:

$$\prod_t P_\theta(a'_t = a_t|S, c_{[0,t]}, a_{[0,t-1]}) \tag{9}$$

This variant investigates whether the supervision of the intermediate actions helps.

**Offline RL**   Lastly, because each turn the agent action affects how the user responds, MUG can be formulated as a RL problem where the user and the UI constitute the environment. We use the Decision Transformer (Chen et al., 2021) for offline RL, by inserting extra learnable return tokens $w_t$ to the $T_{\text{dec}}$ before each action: $T_{\text{dec}}(v, c_0, w_0, v^{a_0}, ..., c_t, w_t)$. The model is:

$$\prod_t P_\theta(a'_t = a_t|S, c_{[0,t]}, w_{[0,t]}, a_{[0,t-1]}) \tag{10}$$

Possible discrete return tokens are $\{1, 2, 3, 4\}$ where 1 on the last turn. During testing, we follow Chen et al. (2021) to force the current turn to have return 1 and adjust prior returns.

| | Challenging | | | | | | All | | | | | |
|---|---|---|---|---|---|---|---|---|---|---|---|---|
| Model | $F1_0$ | $F1_1$ | $F1_2$ | $F1_3$ | $F1_4$ | $avg_{std}$ | $F1_0$ | $F1_1$ | $F1_2$ | $F1_3$ | $F1_4$ | $avg_{std}$ |
| Single | **26.8** | 44.7 | 45.6 | 45.7 | 45.7 | $46.1_{1.3}$ | 56.9 | 60.5 | 60.7 | 60.7 | 60.7 | $60.3_{0.8}$ |
| Ins-only | 25.2 | 49.7 | 52.1 | 52.2 | 52.2 | $53.5_{1.3}$ | 58.5 | 63.4 | 63.8 | 63.8 | 63.9 | $64.0_{0.5}$ |
| Multi | 25.2 | 54.2 | 57.2 | 57.4 | 57.4 | $\mathbf{59.9}_{1.5}$ | **58.6** | **64.3** | **64.9** | **64.9** | **64.9** | $\mathbf{65.1}_{0.2}$ |
| Imitation | 23.5 | **56.5** | **59.6** | **59.6** | **59.6** | $59.4_{1.5}$ | 56.6 | 63.1 | 63.7 | 63.7 | 63.7 | $64.0_{0.8}$ |
| Offline RL | 24.2 | 55.4 | 58.1 | 58.2 | 58.2 | $58.1_{1.1}$ | 58.0 | 64.2 | 64.7 | 64.8 | 64.8 | $\mathbf{65.1}_{0.5}$ |

Table 4: Offline agent F1↑ on the test set. $F1_{0\text{-}4}$ are from model trained with seed 1 and $avg_{std}$ is $F1_4$ of 5 runs.

## 6.2 USER MODELS

Here, we design a simple and effective heuristics-based user model, and then develop a neural version. To show automatic online evaluation is a promising direction for MUG, we also conducted human evaluation on a shared set of 500 examples from the test split (Sec. 6.7).

**Heuristics-based Model**    We observe that, when the selection $a'$ is incorrect, we can deterministically devise a follow-up instruction by using a template as below:

Not the $a'_t$, click the $g$ to/on the *dir*.

This template is to be instantiated on view hierarchy features (in Appx. D). Compared to human follow-ups, heuristic ones are more specific and longer, such as:

- Not the *icon*, click the *action notifications* on the *top right of the screen*.
- Not the *text*, click the *input search* to the *slight right and below of your choice*.

**Neural Instruction Model**    We extend the *Multi* agent architecture to model follow-up commands:

$$P_\phi(c'_t = c_t | S, g, c_{[0,t-1]}, a_{[0,t-1]}) \tag{11}$$

which has decoder $T_{\text{dec}}(v, v^g, c_0, v^{a_0}, c_1, ..., v^{a_{t-1}})$ at turn $t$. For training, we teacher-force at each turn ($t > 0$). We found that using heuristics as prompt greatly boosts development CIDEr (Vedantam et al., 2015) to from 70 to 78. For inference, we use greedy decoding with a maximum length of 12.

## 6.3 METRICS

We focus on evaluating the agent model as it is the pivot task of MUG. Intuitively, we would want the agent to take the desired action $g$ with as few turns as possible. That is,

$$F1_t = \sum_t P(a_t = g | S, c_{[0,t]}, a_{[0,t-1]}) \tag{12}$$

where, in practice, we compute $F1_t$ with early stop over turns to avoid double counting. Clearly, an agent with high F1 and a lower value of $t$ is better than an agent that requires more turns for the same accuracy. With $t$ limited to 0, the task is reduced to a one-pass grounding task.

In an extreme case, we consider an agent with high $F1_0$ but flat changes in $F1_{t > 0}$ to be problematic, since it questions the agent's understanding about the interface. For more comprehensive testing, we also use a simple robustness metric for prediction changes across turns:

$$\Gamma = P(|\{a_t\}| \neq T) \tag{13}$$

which is the percentage of examples that have duplicate actions within $T$ valid turns. We observe that user models, unlike human annotators, sometimes give the same instruction across turns, especially when the agent repeats the same error. Therefore, we only count agent selections that are associated with unique instructions in each example, and ignore other turns.

## 6.4 OFFLINE RESULTS

Tab. 4 presents offline results on the test set, over the *Challenging* (see Sec. 4) and the *All* sets. During inference, we use instructions from the human user and actions from the human agent for turns in between and ask an agent model to predict at each turn. Doing so requires agent models to correct human agent actions, instead of the model's own. Clearly, the models that take into account interaction history outperform those use none or partially. While the *Ins-only* and the *Imitation* models perform closely on the *All* set, they bear larger margins on the *Challenging* and online tests.

| | Model | Heuristics | | | | | | Neural | | | | | |
|---|---|---|---|---|---|---|---|---|---|---|---|---|---|
| | | $F1_0$ | $F1_1$ | $F1_2$ | $F1_3$ | $F1_4$ | $avg_{std}$ | $F1_0$ | $F1_1$ | $F1_2$ | $F1_3$ | $F1_4$ | $avg_{std}$ |
| Challenging | Single | **26.8** | 39.8 | 43.3 | 44.6 | 44.6 | $44.1_{0.5}$ | **26.8** | 41.7 | 43.9 | 44.6 | 45.2 | $44.9_{1.0}$ |
| | Ins-only | 25.2 | 47.4 | 51.7 | 52.9 | 53.5 | $52.9_{1.4}$ | 25.2 | 43.4 | 46.5 | 48.2 | 48.5 | $49.1_{0.7}$ |
| | Multi | 25.2 | **47.8** | 50.9 | 51.7 | 52.4 | $54.3_{1.1}$ | 25.2 | 43.9 | 47.4 | 48.9 | 49.4 | $50.0_{1.1}$ |
| | Imitation | 23.5 | 39.8 | 43.3 | 46.8 | 48.1 | $\mathbf{55.2}_{0.4}$ | 23.5 | 44.1 | **51.4** | **55.5** | **57.6** | $\mathbf{57.7}_{1.5}$ |
| | Offline RL | 24.2 | 47.6 | **52.7** | **54.1** | **54.6** | $54.6_{1.2}$ | 24.2 | **44.6** | 49.4 | 51.3 | 52.0 | $53.4_{1.3}$ |
| All | Single | 56.9 | 65.2 | 67.4 | 68.1 | 68.1 | $68.7_{0.8}$ | 56.9 | 65.0 | 66.5 | 67.0 | 67.4 | $67.1_{0.8}$ |
| | Ins-only | 58.5 | 70.9 | 72.9 | 73.6 | 74.0 | $73.5_{0.4}$ | 58.5 | 67.8 | 69.9 | 70.9 | 71.3 | $70.9_{0.3}$ |
| | Multi | **58.6** | **71.7** | 72.9 | 73.3 | 73.6 | $74.2_{0.5}$ | **58.6** | 67.9 | 69.8 | 70.6 | 70.8 | $71.1_{0.6}$ |
| | Imitation | 56.6 | 69.1 | 72.4 | 73.5 | 73.9 | $\mathbf{74.6}_{0.5}$ | 56.6 | **68.7** | **72.6** | **74.4** | **75.5** | $\mathbf{75.4}_{0.5}$ |
| | Offline RL | 58.0 | 71.6 | **74.0** | **74.7** | **75.0** | $\mathbf{74.6}_{0.6}$ | 58.0 | 68.4 | 71.2 | 72.2 | 72.7 | $73.3_{0.5}$ |

Table 5: Online agent F1↑ on the test set. $F1_{0-4}$ are from model trained with seed 1 and $avg_{std}$ is $F1_4$ of 5 runs.

## 6.5 ONLINE RESULTS

Tab. 5 presents online test scores. In general, models that are supervised by action sequences (i.e., *Imitation* and *Offline RL*) perform better. Both heuristics-based and neural user models are able to guide agents towards task completion. Comparing *Single*'s $F1_0$ and *Imitation*'s $F1_4$, we see that properly using interaction boosts task completion by 18 and 31 on the *Challenging* and *All* test sets.

The average $F1_4$'s show that heuristics-based user works better, except that the *Imitation* collaborates better with the neural user. This might be attributed to the neural user is trained to mimic human command patterns which can be ambiguous and short, while heuristics are more precise while being artificial. This also implies that a large room for further improvement to the user modeling.

Overall, we can see interactive grounding is a challenging task, even on a single screen. The agent modeling involves robust multimodal understanding to self-correct. The user modeling requires controlled language generation, which is still an open problem. The best task completion rate on the *Challenging* subset is only $\sim 55\%$, suggesting a large room for future improvements.

## 6.6 AGENT ROBUSTNESS

We take a deeper look at agent behavior in Tab. 6. We observe that agents with higher F1 tend to be more robust (lower $\Gamma$). The best agent model (*Imitation*) repeats the same mistake for only $16.8\%$ on the *All* test set. However, if we ignore those examples finished in 1 turn i.e., $T > 1$ columns, the repeating rate rises to $\sim 40\%$. The *Heuristics* user, while generally improves agent F1 more than the *Neural* user, has a mixed robustness impact on the *Imitation* and *Offline RL* agents. On weaker agents (the first 3 rows), the *Heuristics* user leads to more salient robustness. These observations suggest improving agent F1 has a more direct and positive impact on robustness.

| | Challenging | | All | | Challenging (T>1) | | All (T>1) | |
|---|---|---|---|---|---|---|---|---|
| | Heuristics | Neural | Heuristics | Neural | Heuristics | Neural | Heuristics | Neural |
| Single | $44.4_{0.9}$ | $44.9_{1.1}$ | $25.8_{0.4}$ | $26.9_{0.3}$ | $60.3_{0.9}$ | $61.0_{1.3}$ | $59.1_{0.9}$ | $61.7_{0.5}$ |
| Ins-only | $37.9_{1.4}$ | $40.5_{1.0}$ | $21.2_{0.4}$ | $23.4_{0.3}$ | $51.4_{1.3}$ | $55.0_{0.8}$ | $51.2_{0.5}$ | $56.4_{0.9}$ |
| Multi | $38.3_{1.3}$ | $41.3_{1.0}$ | $21.3_{0.3}$ | $23.8_{0.5}$ | $51.5_{1.8}$ | $55.6_{1.4}$ | $51.3_{0.8}$ | $57.9_{1.0}$ |
| Imitation | $\mathbf{31.0}_{1.2}$ | $\mathbf{28.3}_{1.4}$ | $\mathbf{17.6}_{0.3}$ | $\mathbf{16.8}_{0.5}$ | $\mathbf{40.7}_{1.7}$ | $\mathbf{37.2}_{1.8}$ | $\mathbf{40.7}_{0.5}$ | $\mathbf{38.9}_{1.1}$ |
| Offline RL | $36.4_{1.1}$ | $35.5_{0.8}$ | $19.9_{0.4}$ | $20.5_{0.3}$ | $48.6_{1.0}$ | $47.4_{1.0}$ | $48.0_{0.7}$ | $49.5_{0.8}$ |

Table 6: Agent $\Gamma \downarrow$ on the test split. Results are from 5 random runs. Smaller $\Gamma$ means more robust.

## 6.7 AUTOMATIC EVALUATION V.S. HUMAN EVALUATION

To show automatic online test is a promising surrogate for human-in-the-loop evaluation, we compare *Single* with *Multi*[2] with a group of human annotators (acting as the *user*) (Tab. 7). We ask the user annotators to follow the same annotation interface and guideline in Sec. 4, and let them to use the trained agent model to ground their commands. That is, human plays the user role and a trained

---

[2]We choose these two models as a pilot study since they perform consistently different in all our metrics.

agent model plays the agent role. This setting maximally mimics a realistic situation where a human user guides the agent to locate a target solely using language commands. The results (Tab. 7) are generally consistent with those from the automatic evaluation (Tab. 5). We should also note that such human study is not meant to reflect every minor differences in automatic evaluations.

| Model | $F1_0$ | $F1_1$ | $F1_2$ | $F1_3$ | $F1_4$ | $\Gamma \downarrow$ |
|---|---|---|---|---|---|---|
| Single | **50.0** | 56.4 | 58.2 | 58.4 | 59.4 | 42.6 |
| Multi | 49.6 | **58.4** | **60.4** | **62.2** | **62.6** | **39.4** |

Table 7: Human-in-the-loop evaluation on 500 examples from the *All* test set. Models are trained with seed 1.

### 6.8 ABLATION ON HEURISTICS

To show agent improves from follow-up instructions effectively, instead of overfitting potential artifacts in the dataset, we report our ablation studies in Tab. 8. Specifically, we focus on the heuristics-based user since it offers well-controlled instruction generation. We can see that random heuristics underperform by $\sim 14\%$ and repeating the initial instruction is even worse. The $\Gamma$ scores also suggest that randomly instantiated instructions are less effective in guiding the agent.

| *Multi* | $F1_0$ | $F1_1$ | $F1_2$ | $F1_3$ | $F1_4$ | $avg_{std}$ | $\Gamma \downarrow$ |
|---|---|---|---|---|---|---|---|
| Heuristics | 25.2 | **47.8** | **50.9** | **51.7** | **52.4** | - | **40.0** |
| Random (5 runs) | 25.2 | 32.7 | 34.3 | 34.7 | 35.1 | $35.6_{0.9}$ | $51.6_{1.5}$ |
| Repeat $c_0$ | 25.2 | 29.3 | 30.9 | 31.6 | 32.0 | - | - |

Table 8: Ablation of instructions using heuristics-based user model for the *Multi* agent on the *Challenging* test set. The *Multi* is trained with seed 1. *Random*: randomly instantiated heuristics for $c_{t>0}$ across 5 seeds.

### 6.9 ERROR ANALYSIS

We manually analyze errors from the best agent (*Imitation*). In Tab. 9, we inspect 30 failed development examples (i.e., unfinished after 5 turns) that are subject to the *Neural* user. Due to the role interplay, we also count problematic commands. We observe that the user model sometimes issues repetitive or uninformative instructions starting from the 3rd turn, leading the agent to the same wrong selection. This might be caused by the data sparsity for examples with $\geq 3$ turns.

| | | Agent | | | | User | |
|---|---|---|---|---|---|---|---|
| Incapabilities | text | icon | UI layout | pos/dir | wrong $c_t$ | stale $c_t$ |
| #Example | 6 | 7 | 9 | 7 | 15 | 27 |

Table 9: Major error categories of the *Imitation* model on 30 failed development examples (150 turns). *stale* $c_t$: repetitive/uninformative instruction. Model is trained with random seed 1.

## 7 CONCLUSIONS

In this paper, we presented MUG, a novel and challenging task for multimodal grounding on UI. MUG requires a grounding agent being able to correct its own prediction, and allows a user to guide the agent via natural language instructions. We contribute a new dataset for the task, investigate various modeling options for the agent and developed multiple evaluation strategies including two user models so as to enable automatic online testing. We found that interaction greatly improves grounding accuracy in the UI domain. Our experiments and analyses also suggest large room for grounding performances, even on a seemingly easy single screen task, which calls for future investigation. Our work contributes to the general effort of multimodal language understanding and its robustness by enabling synchronized multi-turn interactivity.

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

## A  LABELING INTERFACE

Fig. 2 presents the user and agent views in our data collection interface. In the user view, the user can send commands in the message box, to instruct the agent to select the target object as highlighted by a red bounding box on the UI screen. On the agent's view, the agent annotator can respond the user request by performing object selection on the UI screen, which has all the clickble objects highlighted. But there is no indication of the target object so the agent annotator has to guess from the user instruction. The agent is not allowed to text back to the user. The agent's current selection is reflected on the UI screen so the user understands how to further instruct the agent. The annotation task is designed based on the eyes-on hands-free situation of mobile interaction.

## B  DETAILS OF THE LABELING TASK

The labelers of the task were native English speakers and had experience using mobile phones. They were trained with a few pilot tasks to get familiar with the task, during which we also improved the labeling interface and the guidelines based on labelers' feedback. The dataset was completed by 30 labelers in 10 batches. The labeling quality was monitored by sampling examples from each batch for manual examination.

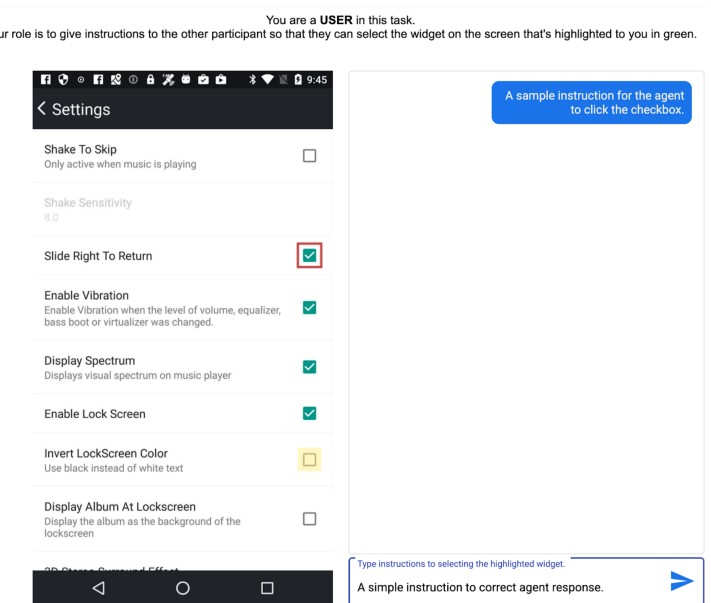

(a) The user sees the target object (boxed in red) and the agent selection in the previous round (boxed in yellow). The user can issue commands in the message box.

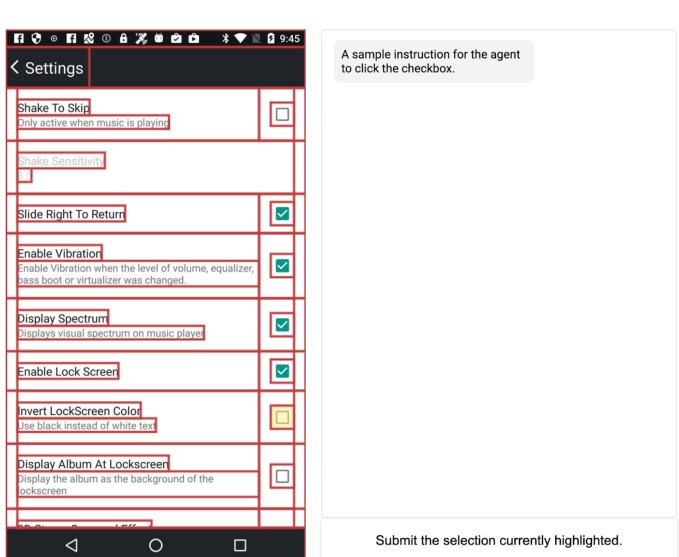

(b) The agent sees the user commands, and all the available candidates (clickable objects) on the screen, which are all boxed in red, and the current selection boxed in yellow.

Figure 2: MUG annotation interfaces consist of a user view and an agent view.

## C  LANGUGAE DIVERSITY

The word-level vocabulary in the training set consists of $13,794$ unique words. Fig. 3 shows the distribution of the 50 most frequent words in the training split with certain non-content words (e.g., *is*, *of*, *comma*) filtered out.

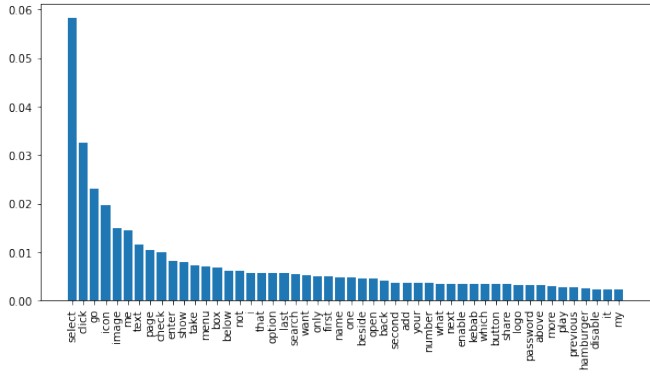

Figure 3: Distribution of top 50 words in MUG training split.

## D   VIEW HIERARCHY FEATURES

Tab. 10 lists the complete view hierarchy features we used. We unify each feature into a real-valued vector. These view hierarchy features are first represented with trainable embeddings, and then encoded by the transformer model (Sec. 6.1). For text attributes (e.g., *text*), we max-pool their non-contextualized token embeddings, which are randomly initialized and trained. For discrete-valued attributes (e.g., *type*), we use a trainable vector for each possible value. The ordering of objects in transformer input follows the pre-order traversal in the view hierarchy (which is a tree structure). We then combine the vision representations of individual UI objects via ROI pooling over ResNet featuremap of the encoded screenshot image, and view hierarchy encoding to form a multimodal representation of each UI object for the downstream computation of the model.

We consider these view hierarchy features to be auxiliary. There is often a huge gap between what command the user would issue based on what they see on the UI, and what the underlying information is for the UI. As we discussed in Sec. 6, about 46% of UI objects do not have a text label, and the user would need to come up with their own language description about the object, which is why the text matching baseline fails. Even when there are text descriptions, they are not necessarily what the user would articulate since a user command can be abstract. Fundamentally, the internal representation of the UI is often inaccessible or uninterpretable to the user, thus calling for the help of multimodal modeling and interaction modeling.

| Feature | Example |
|---|---|
| bounding box | [xmin, xmax, ymin, ymax] |
| leaf | true/false |
| type | button/checkbox/... |
| clickable | true/false |
| text | email address/passcode |
| resource id | login_icon |
| dom | [pre/post-order index] |

Table 10: Features $\psi$ used for visual structure.

## E   HYPERPARAMETERS & TRAINING

For all our agent models, we use the same configurations, which are grid-searched based on models' offline validation performances. Our hyperparameters are chosen from the best offline development F1 scores. For the number of self-attention modules, we grid-searched in $\{1, 2, 4, 6\}$, which resulted in 2 hidden layers for the user interface Transformer encoder and 6 hidden layers for the grounding decoder. Each self-attention module uses 8-head self and encoder-decoder attention with a 256 hidden size. The dropout rate for attention and MLP layers is $0.1$, which is grid-searched in $\{0.1, 0.2, 0.5\}$. For learning rate, we grid-searched from $\{1e{-}3, 3e{-}4, 1e{-}4, 3e{-}5, 1e{-}5\}$, and use $3e{-}4$ with linear warmup with cosine annealing for the first 10k steps. All the models are trained to

100k steps with a batch size of 128 on a 32-core Google Cloud TPUv3. Models are evaluated every 1k steps and the version with the best development offline $F1_4$ is saved. The training time for our agent model is around 8 hours.

Our neural user model has the same grid-searched configuration as the agent, i.e., 2 encoder layers, 6 decoder layers, 0.1 for dropout, and the same warmup scheduling. The best learning rate is $1e{-}4$. Different from the agent model, we found the neural user model's development CIDEr score quickly drops after 6k steps, possibly due to overfitting and data sparsity, thus its training early-stops there.

## F  EXAMPLES IN THE MUG DATASET

We present some examples from the MUG dataset in Fig. 4 and 5. Each example contains instructions and selections from human user and agent annotators.

## G  LIMITATIONS

**English-only Dataset**   While non-English examples exists, we acknowledge that MUG mostly consists of English UI. Other languages do exist in the dataset, but consists of a small portion. Specifically, our instructions are English-only. Future extensions to our work should address or alleviate this issue.

**Platform-specific Interfaces**   Our interfaces, since coming from RICO, only consist of Android screens. In practice, it is also difficult to obtain non-Android interfaces. We acknowledge this is an application limitation. And the bias from the top and bottom banner of Android could make trained model brittle in other domains.

**Going beyond Single Screen**   We aim to establish the task and report baseline performances for future work. The interaction in MUG happens within the same user interface. A natural extension would be extending the task to span over sequence of interfaces. Indeed, the task would become more challenging, and potentially require large offline training data and reliable online simulation.

**Better User Model**   The current best neural instruction generation we use has a CIDEr 78.0 on the validation set. We acknowledge there is space for further improvement. Note that our neural instructions are trained on multi-turn examples in MUG, which amounts to $\sim 20\%$ of the training data. It suggests external resources could be useful for improving user model performances.

**Interaction Dynamics between User and Agent**   It would be helpful to study how/why the agent sometime repeatedly makes incorrect actions in Tab. 6, such as whether repeated mistakes are due to the lack of language utility/diversity in user instruction or the lack of understanding in the agent.

## H  PREDICTION EXAMPLES

Here, we demonstrate predictions from the *Imitation* model. Fig. 6 demonstrates successfully solved examples following the instructions generated by the *Heuristic* user model, while failed ones are in Fig. 7. Similarly, Fig. 8 demonstrates solved ones following the instructions generated by the *Neural* user model, and failed ones are in Fig. 9.

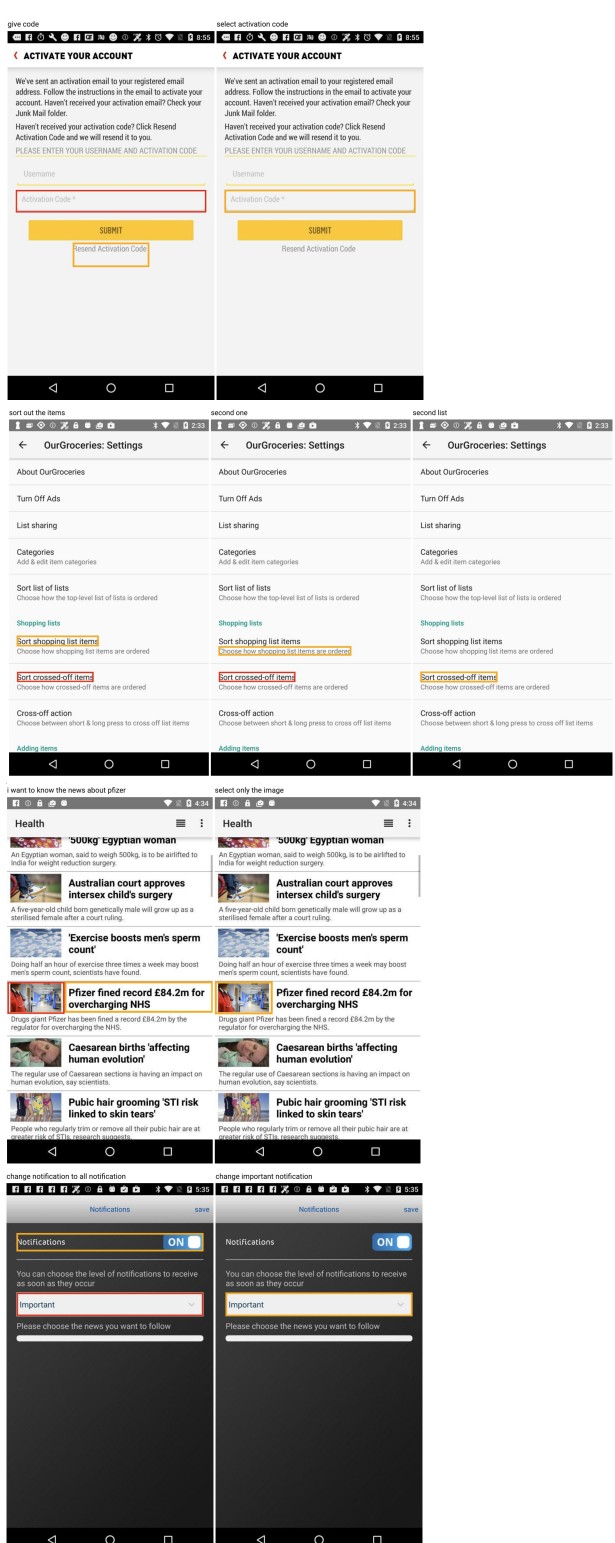

Figure 4: MUG examples 1-4. Instructions are at top of each turn. Agent selection is in ☐ and target is in ☐.

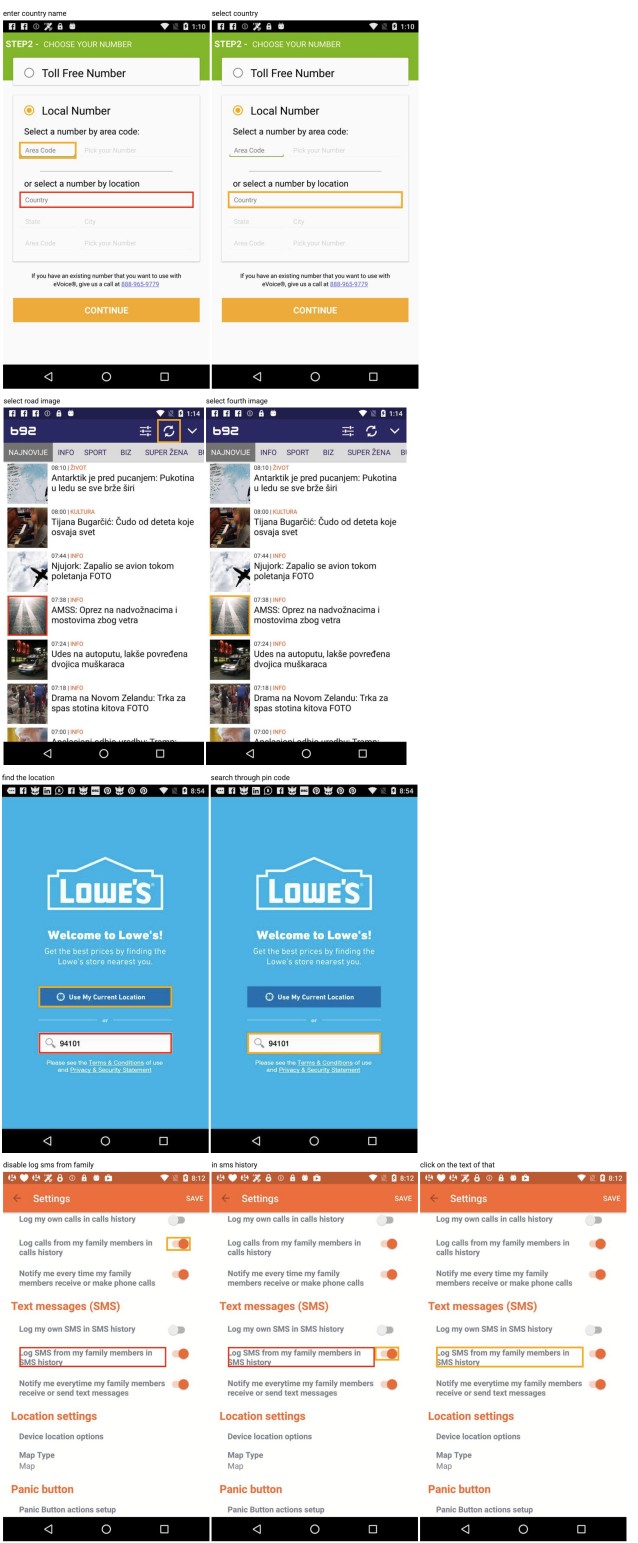

Figure 5: MUG examples 5-8. Instructions are at top of each turn. Agent selection is in ☐ and target is in ☐.

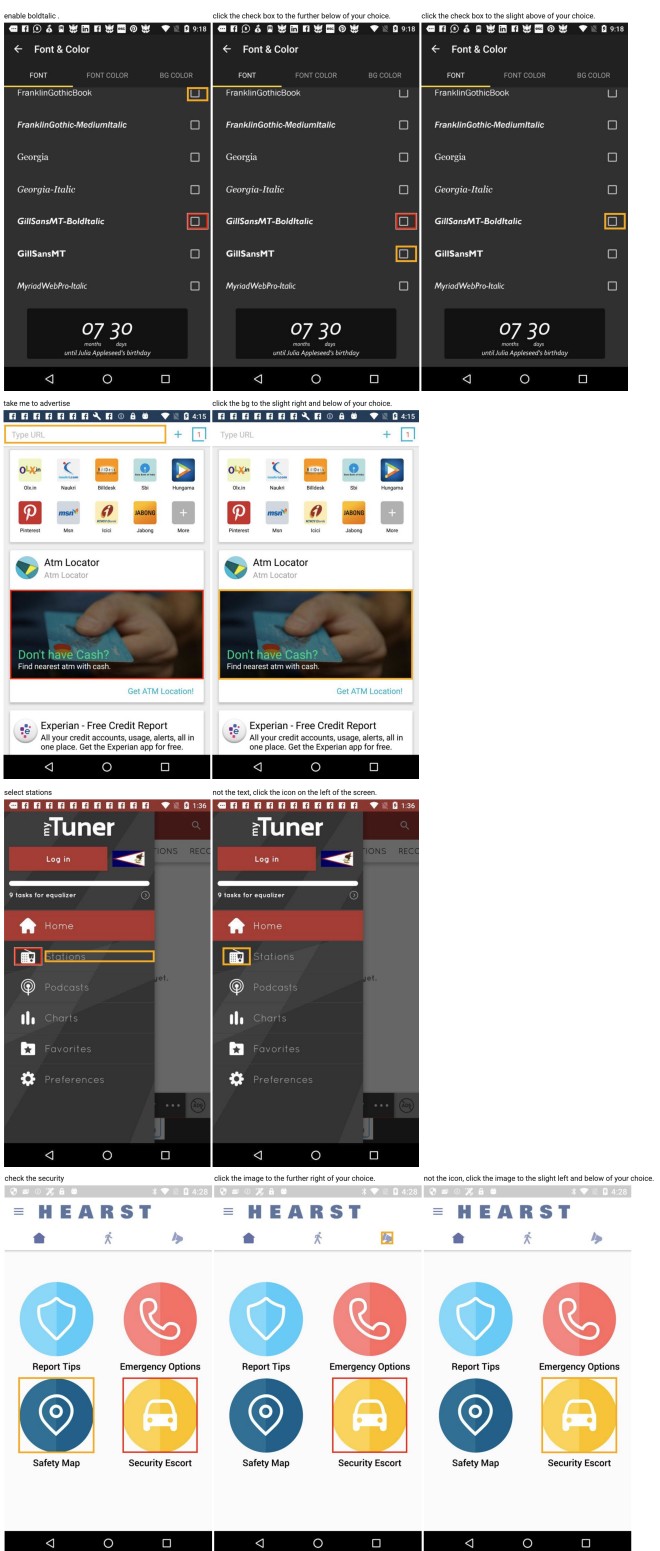

Figure 6: **Completed** examples by the *Imitation* agent following the instructions generated by the **Heuristic** user.

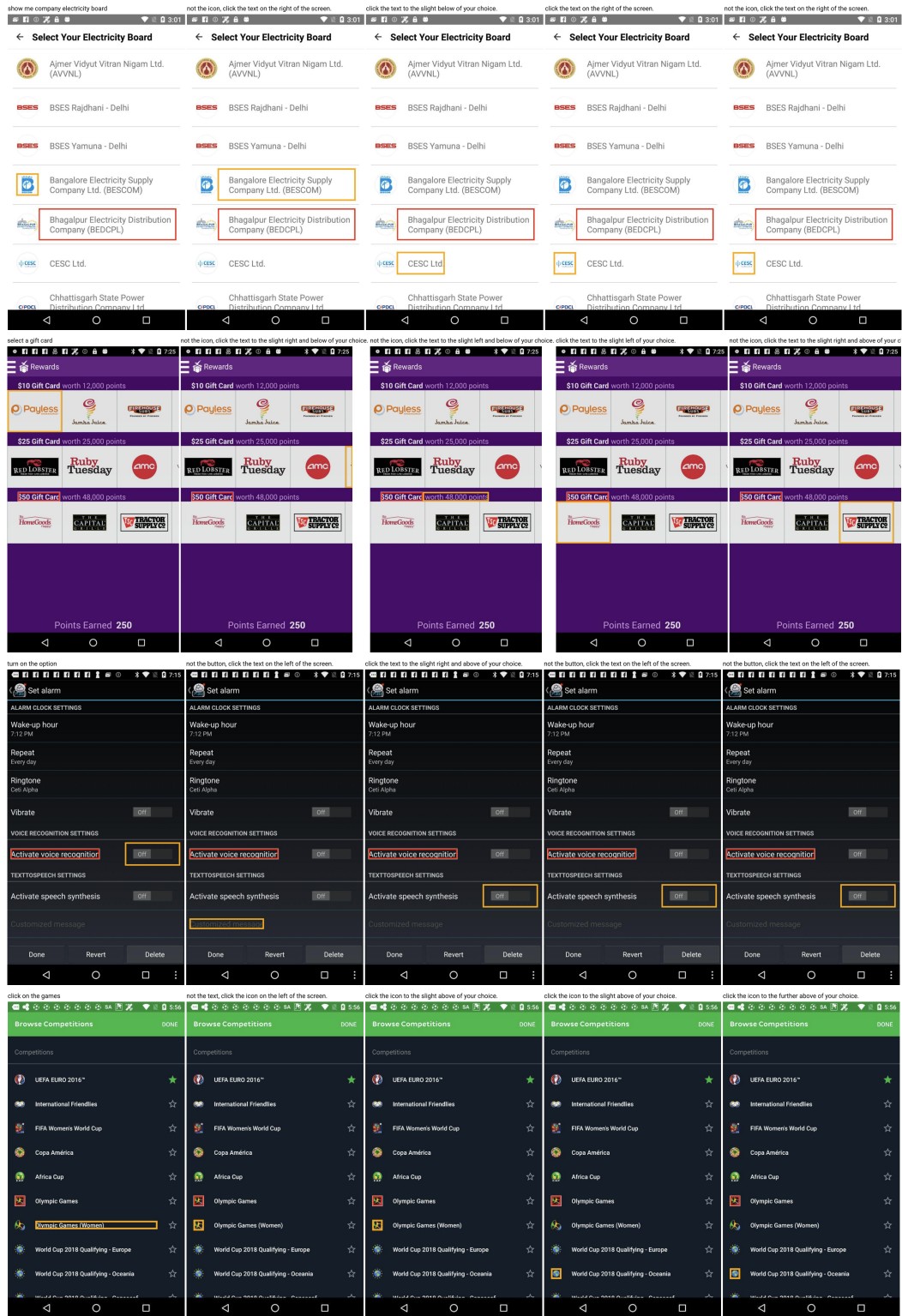

Figure 7: **Failed** examples by the *Imitation* agent following the instructions generated by the **Heuristic** user.

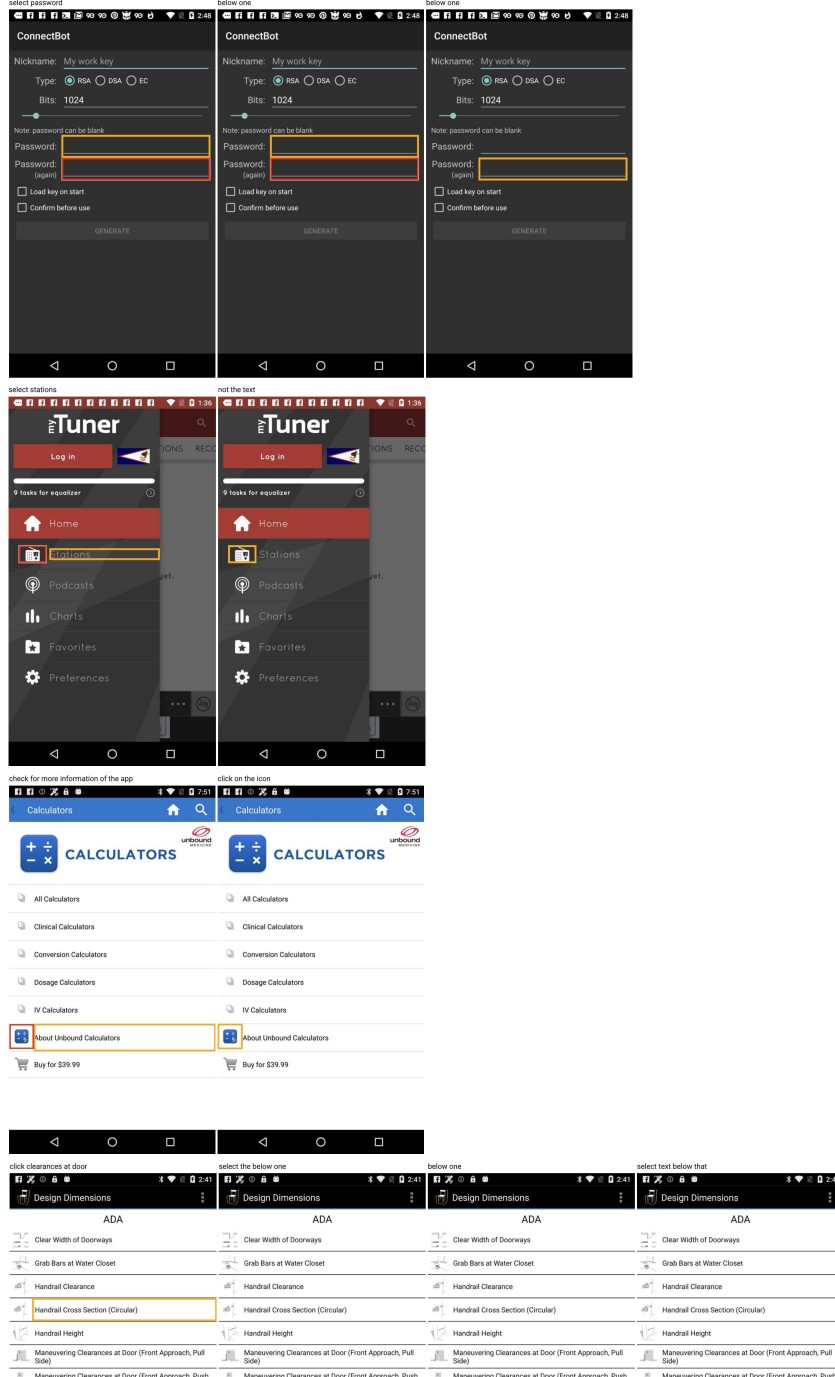

Figure 8: **Completed** examples by the *Imitation* agent following the instructions generated by the **Neural** user.

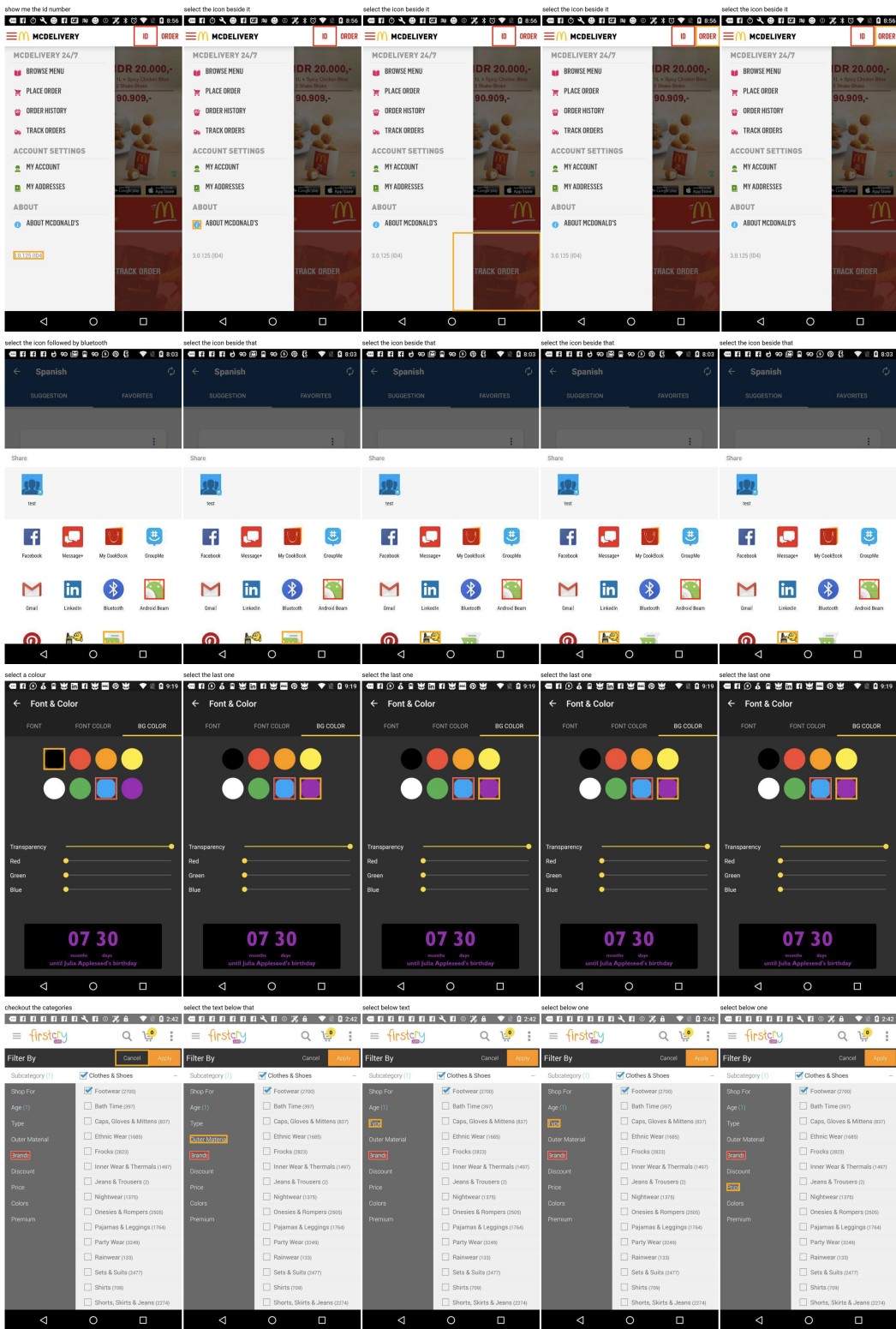

Figure 9: **Failed** examples by the *Imitation* agent following the instructions generated by the **Neural** user.

