# OpenReview forum: "MUG: Interactive Multimodal Grounding on User Interfaces"
_ICLR.cc/2023/Conference — Submitted to ICLR 2023_

### Official Review · Reviewer_tBFu · 2022-10-18

**Confidence:** 4
**Correctness:** 3
**Technical Novelty And Significance:** 2
**Empirical Novelty And Significance:** 2
**Recommendation:** 3

**Clarity, Quality, Novelty And Reproducibility:**

The paper is generally well-presented although important information is glossed over.  In particular, I wasn't sure how the view hierarchy features were encoded and what role the screenshot played.   It also seemed to me that a simple hand-crafted baseline could have done very well on this task.

**Strength And Weaknesses:**

The authors will make both the dataset and the code available.  This dataset will be a useful addition to those working on this type of interaction and more generally for those interested in multi-modal processing systems.   However, the outputs appear to be limited to clicking buttons and check boxes.

**Summary Of The Paper:**

This paper describes a new database for training a multimodal agent in a simple grounding task.  The user is faced with a screen and wishes to select a radio button or checkbox by voice.  If the agent gets it wrong, the user can attempt to correct the agent by a followup command such as "click the checkbox below".  The data was collected using a wizard of oz paradigm in which the human user sees a screen display and the randomly sampled UI object to be selected.  The user then issues an appropriate written (?) command.  The human agent sees the same screen with all clickable objects marked and the user's input.  The agent then selects the most likely target and the process repeats.  All data is derived from an existing data set called RICO, first published in 2017.

In addition to the data, the authors provide benchmark models and experimental results.  The benchmark model consists of a resnet encoder for the screen which is then plugged into a transformer along with the view hierarchy features. The resulting embedding is then input to an decoder along with the previous turns of the "conversation".  The agent is tested using both a heuristic and neural trained user model.


**Summary Of The Review:**

There is little that is original in this paper and I found the motivation for the work unconvincing.  If the goal is to empower a UI agent to respond to natural language commands as well as direct clicks, then why would you complicate matters by conditioning with a resnet encoded screen shot?  If the agent rendered the display, it knows exactly where everything is so why not simply create a list of the available UI objects and their spatial relationship.

---

> ### Author Response · Authors · 2022-11-18
> **Response to Reviewer tBFu**
>
> Thank you for your detailed reviews. We have addressed all these issues in the rebuttal and the revision that we submitted. We hope you can kindly reconsider the score for our paper in light of these improvements. Please let us know if you have any further questions.
>
> > how the view hierarchy features were encoded and what role the screenshot played
>
> We encode the view hierarchy in a similar way to previous work. At the high level, the view hierarchy features are first represented with trainable embeddings, and then encoded by the transformer model. Screenshot is important as it captures the visual appearance of UI elements. As mentioned in our paper, we combine the vision representations of individual UI objects via ROI pooling over ResNet featuremap from the encoded image, and view hierarchy encoding to form a multimodal representation of the UI for the downstream computation of the model. We have clarified the details of view hierarchy encoding and multimodal representation in the appendix (Appx. D) of the revision.
>
> > It also seemed to me that a simple hand-crafted baseline could have done very well on this task.
>
> We have added a simple baseline in our experiments as you suggested that matches user command against UI object text. The deterministic baseline uses METEOR scores which uses stem and lemmatization matching and offers more robust matching than ngram. The baseline only achieved ~21% accuracy. The cause for the low accuracy of text  matching involves two challenges: 1) view hierarchy often lacks text information, e.g. ~46% of UI objects do not have a text feature, thus a method will need to rely on other modalities for grounding); 2) user instructions are often short and abstract, thus need follow-up commands to guide the model. We have added these new results in the Experiment section (Sec. 6).
>
> > There is little that is original in this paper.
>
> Our contributions are on the task aspects rather than modeling. We acknowledge that we used common model architectures in the UI domain, although there are some unique aspects in using these models in our task such as how to represent interaction history. But we want to point out that the major contributions of our work are the introduction and formulation of interactive grounding tasks along with the new dataset and the benchmark results. Corrective interaction in UI grounding is largely overlooked and challenging. We believe this contribution is original, and paves the way for others to further investigate the topic. We have already released our new dataset to the public. We have added extra clarification on the modeling part (Sec. 5).
>
> > If the agent rendered the display, it knows exactly where everything is so why not simply create a list of the available UI objects and their spatial relationship.
>
> There is often a huge gap between what command the user would issue based on what they see on the UI, and what the underlying information is for the UI. As we discussed earlier, about 46% of UI objects do not have a text label, and the user would need to come up with their own language description about the object, which is why the text matching baseline fails. Even when there are text descriptions, they are not necessarily what the user would articulate since a command can be abstract. Fundamentally, the internal representation of the UI is often inaccessible or uninterpretable to the user, thus calling for the help of multimodal and interaction modeling. We have added this explanation in the revision (Sec. 6 and Appx. D) to make it clearer.

---

### Official Review · Reviewer_t7tT · 2022-10-25

**Confidence:** 4
**Correctness:** 3
**Technical Novelty And Significance:** 2
**Empirical Novelty And Significance:** 2
**Recommendation:** 5

**Clarity, Quality, Novelty And Reproducibility:**

The paper proposed to create a large dataset for the interactive GUI Grounding task. It clearly described how the dataset is collected, and also built different model variants and metrics to investigate the relative statistics of the task.


**Strength And Weaknesses:**

The main contribution of this paper is to create a rich dataset which contains the human agent interactions for language grounding on a GUI screen. The paper also built some benchmarks based on different model variants and evaluation strategies to show the interaction  significantly improves grounding accuracy.
Some questions:
1. Besides the Mobile UI grounding task, there are also some tasks in interactive grounding, for example in the HRI field, what’s the difference between GUI Grounding and other fields’ interactive grounding task?
2. What’s the difference between online results and offline results?
3. It’s not clear how you evaluate the human-in-the-loop setting?


**Summary Of The Paper:**

In  this paper,  it proposed a novel interactive task for multimodal grounding where a user and an agent work collaboratively on an interface screen.  The authors collected a new large dataset for interactions between human and agent, in which 20% are more than 1 round. The paper also experimented with different modeling variants and evaluation strategies to build different benchmarks on different statistics.


**Summary Of The Review:**

This paper clearly presents and verifies the contributions including the dataset created and the experimental benchmarks.  As it mainly focuses on the interactions and grounding. It might be better to submit to nlp related conferences or dialog related conferences.

---

> ### Author Response · Authors · 2022-11-18
> **Response to Reviewer t7tT**
>
> Thank you for your careful reviews. We have clarified all your questions in the revision, which makes the paper stronger. Please let us know if you have further questions.
>
> > Besides the Mobile UI grounding task, there are also some tasks in interactive grounding, for example in the HRI field, what’s the difference between GUI Grounding and other fields’ interactive grounding task?
>
> HRI and UI grounding are two different domains. Our understanding is that the former aligns entities (either in simulated environments or real-world visions) to language. The latter aligns entities on the UI screen. Both domains require the agent to understand the implicit semantics of entities and the environment, and thus have different challenges in their domains.
>
> > What’s the difference between online results and offline results?
>
> In our paper, online results denote that we use a user model (instead of a human user) to emit follow-up instructions based on agent inferences at each turn. This is different from offline results which use recorded human-agent interaction traces and ignore intermediate agent predictions.
>
> > It’s not clear how you evaluate the human-in-the-loop setting?
>
> For human-in-the-loop evaluation, we re-use the same annotation interface and guideline, and ask the human labeler to use the trained agent model to ground their commands. This setting maximally mimics a realistic situation where a human user guides the agent to locate a target solely using language commands. In this setting, the user role is played by human and the agent role is played by a trained model. We have clarified this in sec 6.7 in the revised version.

---

### Official Review · Reviewer_YoBY · 2022-10-25

**Confidence:** 4
**Correctness:** 3
**Technical Novelty And Significance:** 3
**Empirical Novelty And Significance:** 3
**Recommendation:** 5

**Clarity, Quality, Novelty And Reproducibility:**

The paper is largely clear with clear figures and exposition. It was a joy reading this paper. The paper makes important contributions to the data collection and model evaluation, with important insights for future work.

**Strength And Weaknesses:**

Strengths:
1. The paper is well motivated and described. The ideas are clear and experiments are on several large multimodal datasets.
2. The paper is largely clear with clear figures and exposition. It was a joy reading this paper.
3. The paper makes important contributions to the data collection and model evaluation, with important insights for future work.

Weaknesses:
1. There can be more comparisons and evaluation of actual multimodal methods - right now all the tested methods are largely on interactive learning/online and offline RL, but the paper is motivated as a multimodal paper. See https://arxiv.org/abs/2107.07502 for an example of a recent multimodal benchmark and a suite of tested multimodal fusion models with key takeaways for future work, and see https://arxiv.org/abs/2209.03430 for more references in the field of multimodal grounding and the key methods there.
2. There can also be a better motivation and definition of multimodal grounding in this task - is grounding simply finding the right region of interest as a bounding box? How does grounding change across interaction? How important is grounding to the specific object versus whole UI-level representation? What are the effects of the amount of data required for grounding? Many questions should be formalized or motivated here. Given that this seems to be the main selling point of the dataset there needs much more analysis on how current grounding (or more generally alignment) methods work on this task and what future innovations are needed.

**Summary Of The Paper:**

This paper presents present MUG, a novel interactive task for multimodal grounding where a user and an agent work collaboratively on an interface screen, essentially bridging multimodal applications with user interfaces in HCI research. Their main novelty is an interactive task as opposed to existing static interaction tasks in user interfaces, and multiple rounds of interactions enables the user to give further commands for the agent to refine or correct its actions. This paper presents a dataset of 77,820 sequences of human user-agent interaction on mobile interfaces to study this problem and experiment several models and evaluation strategies, finding that iterative interaction improves task completion accuracy but also presents directions for future work.

**Summary Of The Review:**

There can be more comparisons and evaluation of actual multimodal methods, including better motivation and definition of multimodal grounding in this task.

---

> ### Author Response · Authors · 2022-11-18
> **Response to Reviewer YoBY**
>
> Thank you very much for your insightful feedback. We have released our dataset to the public. Please let us know if you have further questions.
>
> > There can be more comparisons and evaluation of actual multimodal methods - right now all the tested methods are largely on interactive learning/online and offline RL, but the paper is motivated as a multimodal paper.
>
> We focus on the interaction aspect of multimodal tasks. Our aim is to establish the task and present baseline results from common architectures in the UI domain, rather than proposing new multimodal modeling. The concrete multimodal fusion options are vast, but not the focus of this work. We have further clarified our focus (Sec. 5) on multi-turn aspects in the revision.
>
> > There can also be a better motivation and definition of multimodal grounding in this task - is grounding simply finding the right region of interest as a bounding box? How does grounding change across interaction? How important is grounding to the specific object versus whole UI-level representation? What are the effects of the amount of data required for grounding? Many questions should be formalized or motivated here. Given that this seems to be the main selling point of the dataset there needs much more analysis on how current grounding (or more generally alignment) methods work on this task and what future innovations are needed.
>
> Multimodal grounding in  UI is already a well-established task. Our motivation is to broaden existing multi-modal tasks and make it interactive and more realistic.
> - >Is grounding simply finding the right region of interest as a bounding box?
>
>     Grounding can have different formulations in different domains. For grounding in UI, it involves clicking on the interface. Clicking is often modeled as a classification problem over objects which are visually represented by the region contained in the box. Our task formulation does not reinvent this aspect.
>
> - >How does grounding change across interaction?
>
>     This is what we focus on in this paper. As an initial work, we want to know the impact on the grounding performances. As shown in our experiment section, the grounding performances gradually increase over interaction turns.
>
> - >How important is grounding to the specific object versus whole UI-level representation?
>
>     The vision modality of specific objects are ROI-pooled from the entire UI image. This modality is essential since ~46% objects do not have proper text features.
>
> - >What are the effects of the amount of data required for grounding?
>
>     In this paper, we showed an important observation that adding interaction data to non-interactive grounding data boosts task completion by a large margin. Therefore, we believe large and clean data naturally gives a positive impact on grounding performances.
>
> Simple text alignment method does not work due to the incomplete text features in view hierarchy. Generally speaking, the agent architecture we used is also an alignment model as it classifies which object best aligns to the interaction history given the screen context. As to takeaway for future works, we have pointed out limitations and our conjectures in the experiments, analysis, and Appx. G.

---

### Official Review · Reviewer_FLkv · 2022-11-01

**Confidence:** 4
**Correctness:** 3
**Technical Novelty And Significance:** 2
**Empirical Novelty And Significance:** 2
**Recommendation:** 5

**Clarity, Quality, Novelty And Reproducibility:**

In general this paper was a joy to read, with very digestible takeaways. The novelty of this dataset with respect to prior work is clear (though perhaps a bit understated), and I’m excited to see this dataset and baselines released as mentioned in the work!


**Strength And Weaknesses:**

I believe this to be a well constructed dataset, with applications for assistive technology, and a nice playground for studying deeper questions of multimodal language grounding.

However, I do believe this work has several weaknesses; while the paper focuses on the multi-turn aspect of their data collection, only 20% of the collected interactions actually require “revisions” or “corrections” from the human — this seems rather low for a dataset that hopes to spawn future work in multi-turn interactions. Furthermore, the type and diversity of language that human users give is rather limited; as noted in the work, most utterances are only ~4 words long; it’s not clear the data is rich enough to enable multi-turn grounding research, or if the difficulty is just in mapping “key phrases” to UI components — it would have been nice to see this ablation.

Finally, the choice of Offline RL as an ablation seems a bit unnatural; given that we know the “right” UI component to select in each interaction, treating this as a multi-step MDP doesn’t feel appropriate; instead, it’s almost as if something like DAgger would be the better baseline —> this would also be strictly better than the imitation learning baseline — I would love to see how much this improves on the existing results.


**Summary Of The Paper:**

This work proposes a novel benchmark dataset for language grounding on user interfaces, notably exploring the ability to handle *sequences* or multiple turns of instructions/actions between a human user and virtual assistant. This dataset is of remarkable size, consisting of 77K unique language utterances, grounded in 31K individual mobile app “screens” belonging to a set of 7K unique mobile applications. In addition to the dataset itself, this work proposes a straightforward baseline with a set of meaningful ablations that characterize the difficulty and open challenges of working on this task.

The dataset is collected by pairing two human annotators together, with one human acting as the “speaker” and the other as the “follower” — similar to many other multi-turn data collection procedures in the NLP literature. In addition to assembling this dataset, the work further splits the data into different splits based on the “challenge” level of the interactions in question — where “challenge” is a function of taking more than 1 turn to get to the correct “answer” (agent interaction on the screen). That means that ~80% of the dataset falls into the single-turn instruction following paradigm.

The proposed approach pairs an object-oriented visual tokenizer with a Transformer decoder that eats the history of state/actions and outputs the a “context” vector that is used to score each object to interact with (e.g., as a ranking problem). The ablations of the approach show that single-turn instruction following is not enough, and that modeling the entire context, and ability to make decisions in sequence (e.g., via imitation learning) is critical to performing well on this dataset.


**Summary Of The Review:**

This is a decent dataset paper, proposing an interesting resource for language grounding on user interfaces. However, I don’t believe the paper as written delivers on its promises; I think the language diversity is lacking, and the multi-turn nature of the dataset is a bit oversold.

I would really love to see an ablation experiment that just maps key phrases (bigrams, trigrams) to corresponding UI components to see if the “difficulty” of this dataset is actually multi-turn grounding, or just having the right mapping of words to components on the interface —> if it’s the latter, then it feels like any pretrained model with some amount of knowledge of UI/web components (e.g., WebGPT) would make this dataset trivial.

---

> ### Author Response · Authors · 2022-11-18
> **Response to Reviewer FLkv**
>
> Thank you for your thorough comments and constructive feedback. We have released our dataset. We address each of your questions here and in the revision. Please let us know if you have further questions.
>
>
> > only 20% of the collected interactions actually require “revisions” or “corrections” from the human —...this seems rather low for a dataset that hopes to spawn future work in multi-turn interactions
>
> We argue that the 20% examples that need revision leads to a large impact in practice.
> It lines up an upper bound for agent model performances (i.e., 80 F1 on initial instructions). In practice, such as in our experiments, agent models perform much worse than that. Less than 60% examples are solved within one turn.
> We should note that the 20% is for single-screen cases. Real-world grounding sessions can also span multiple screens. If we apply the same 20% rate here, then the probability for a multi-screen session to require revision is much more significant. For instance, with 5 screens, the probability goes to ~67%.
> We have added these to the revision (Sec. 4.2) to strengthen our motivation.
>
> > multi-turn nature is oversold
>
> Our work is an initial one along this line. Revisions are needed mostly when an agent makes errors or user specified ambiguous commands. This is a scenario overlooked in prior works. And once it happens, the task becomes much more challenging than a traditional grounding task.
>
> > I would really love to see an ablation experiment that just maps key phrases (bigrams, trigrams) to corresponding UI components to see if the “difficulty” of this dataset is actually multi-turn grounding, or just having the right mapping of words to components on the interface —> if it’s the latter, then it feels like any pretrained model with some amount of knowledge of UI/web components (e.g., WebGPT) would make this dataset trivial.
>
> We built a simple matching baseline (based on METEOR score which is supposedly better than ngram since it uses stem and lemmatization matching) that matches UI object text features and user instructions. The F1@0 is only ~21%. Simply matching text information involves two challenges: 1) view hierarchy is often incomplete in UI dataset (e.g. ~46% of UI objects do not have a dedicated text property in RICO, thus will need other modalities); 2) user instructions are often short and ambiguous, thus need follow-up commands. We have clarified this in the experiment section (Sec. 6) in the revised version.
>
> > the type and diversity of language that human users give is rather limited; as noted in the work, most utterances are only ~4 words long; it’s not clear the data is rich enough to enable multi-turn grounding research.
>
> Individual instructions tend to be short in realistic use cases. Initial instructions and their follow-ups form a more challenging context. This is in contrast to prior works that aim to increase the language utility of follow-up instructions, making them more independent between turns and thus less realistic. We should note that the task is challenging even though the commands are short. To have a concrete presentation, we have added additional stats in the appendix (Appx. C). The unfiltered vocab size at word-level is 13,497 in the training set. And we added a plot of the top-50 frequent words, to give a sense about the distribution.
>
> > instead, it’s almost as if something like DAgger would be the better baseline
>
> Using DAgger would involve querying the user model (as a surrogate to a real expert) during the training. However, we are also using the same user model for online evaluation, thus doing so runs the risk of overfitting the evaluation user. Therefore, we kept the user model separate from agent training to avoid leaking test information.

---

### Author Response · Authors · 2022-11-29
**Regarding Rebuttal**

Dear reviewers, we have updated the draft and responded your questions. Hopefully our response addresses some of your concerns. Please check it out and let us know if you have any questions. Would be more than happy to discuss.

---

### Decision · Program_Chairs · 2023-01-20

**Decision:**

Reject

**Justification For Why Not Higher Score:**

The task is simple: just click buttons and check boxes. About 80% of the interactions are single turn and most sentences are only 4 words. Comparisons with similar work are needed.  The selection of baseline models could be improved.

**Justification For Why Not Lower Score:**

NA

**Metareview: Summary, Strengths And Weaknesses:**

Summary:

This paper presents a new dataset and a benchmank for language grounding on user interface, where a user and an agent collaboratively on an interface screen.  This dataset is quite large in which 20% are more than one round. For data collection, two humans are working together using a wizard of oz paradigm to complete certain tasks.  Experiment are conducted with multiple models and evaluation strategies, finding that iterative interaction improves task completion accuracy but also presents directions for future work.


Strength:

The paper is well motivated and described. The main contribution is the dataset which contains the human agent interactions grounding on a user interface screen, as well as multiple baseline models showing the interaction can truely improves grounding accuracy.   Both the dataset and the codes will be released publicly.


Weaknesses:

The task is simple: just click buttons and check boxes. About 80% of the interactions are single turn and most sentences are only 4 words. Comparisons with similar work are needed.  The selection of baseline models could be improved.

**Summary Of Ac-Reviewer Meeting:**

NA